## Review/Meta-analysis

suicide; self harm; effectiveness; Global mental health delivery; interventions

**Corresponding author:**
Charles Zemp;
Email: zempc@tcd.ie

# Self-harm and suicide prevention in humanitarian and fragile contexts: A systematic scoping review

Charles Zemp[1] ⓘ, Frédérique Vallières[1], Fabian Broecker[1],
Emily E. Edmunds Haroz[2] ⓘ, Isabella Kakish[1], Greg Sheaf[3], Joshua Sung Young Lee[4],
Sarah Harrison[4] and Rikke Siersbaek[5]

[1]Trinity Centre for Global Health, Trinity College Dublin, Ireland; [2]International Health, Mental Health, Johns Hopkins University Bloomberg School of Public Health, USA; [3]The Library of Trinity College Dublin, Ireland; [4]Red Cross Red Crescent Movement MHPSS Hub, Denmark and [5]Trinity College Dublin School of Medicine, Ireland

## Abstract

Suicide remains one of the leading causes of death globally, with growing evidence that humanitarian emergencies and fragile states, most of which unfold in low- to middle-income countries (LMICs), are associated with elevated risk of suicide. However, the few suicide-targeted interventions for use in humanitarian contexts remain both sparse and fragmented. This scoping review aims to identify and synthesise evidence from suicide and self-harm prevention interventions implemented in all types of humanitarian settings, globally, that have been evaluated for their effectiveness in improving suicide and self-harm-related outcomes. We systematically searched eight electronic databases, including two grey literature databases, and relevant organisational websites for records published through November 2024 and in any language. Screening was done using the Covidence platform, with each record independently screened by two reviewers. Among other preselected inclusion criteria, studies must have conducted a quantitative evaluation of the effectiveness of an intervention on improving suicide and self-harm-related outcomes during a humanitarian crisis to be included for data extraction. Data extraction and quality assessment were both conducted by two authors. In all, 6,209 records were screened at the title and abstract phase; 104 were included for full text screening; and 23 studies were included for data extraction. Most studies were conducted during the coronavirus disease 2019 pandemic (COVID-19), and in high-income countries. Evaluated interventions encompassed various approaches, including psychotherapeutic, practical, and pharmacological assistance, often employing multiple components. The majority targeted the general population, were delivered via remote modalities and relied on mental health specialists for their administration. Overall, 15 (65.2%) interventions were associated with statistically significant positive effects on suicide and or self-harm-related outcomes. Promising approaches include cognitive behavioural therapy-based text services, skills-building programmes, and strategies that foster supportive environments for high-risk individuals. These findings highlight both promising approaches and critical gaps in suicide prevention efforts in humanitarian settings. The limited evidence base – particularly in LMICs and with particularly at-risk populations – alongside the increasing frequency of humanitarian crises, underscores the urgent need for future implementation and associated research of suicide and self-harm prevention initiatives within humanitarian contexts.

## Impact statement

Suicide and self-harm are both pressing concerns within global mental health, with prevalence rates remaining high despite significant reductions in the global suicide mortality rate over the past three decades. Humanitarian crises – such as natural disasters, armed conflicts, forced displacement and public health emergencies – are known to increase the risk of suicide and/or self-harm thoughts and behaviours. Although suicide and self-harm are both preventable through evidence-based interventions, suicide prevention has only recently begun to receive dedicated attention within humanitarian programming. Previous reviews have assessed the effectiveness of interventions targeting suicide and self-harm in humanitarian contexts, but these have been limited to specific types of emergencies. In our review, we synthesise the global evidence base on suicide and self-harm prevention interventions across all types of humanitarian and fragile settings, assessing intervention effectiveness in improving suicide and/or self-harm outcomes. In doing so, we not only highlight a selection of promising approaches but also significant gaps in the evidence base for suicide prevention in humanitarian crises, most of which occur in low- to middle-income countries. Our findings have direct implications for





strengthening suicide prevention efforts in humanitarian contexts, and we provide recommendations to guide future empirical work and resource development. Ultimately, the results of our review lay the groundwork for the development of robust, evidence-informed practical guidance to help frontline humanitarian workers respond more effectively to suicide and self-harm risk in the field.

## Introduction

Suicide remains a major global public health crisis, claiming over 720,000 lives each year (WHO, 2025). The global prevalence of 'self-harm' or 'non-suicidal self-injury' (NSSI), a strong predictor of suicidal behaviour, is 17.7% (Moloney et al., 2024). Notably, these statistics almost certainly underestimate the true burden of suicide, as widespread stigma and legal, religious and cultural prohibitions against suicide, as well as poor or absent suicide surveillance in many countries, account for the considerable under-reporting of suicide-related deaths (WHO, 2025). Moreover, the ramifications of suicide extend far beyond the individual (Knipe et al., 2022), with surviving family members and friends commonly experiencing prolonged grief marked by guilt, shame, despair and a heightened risk of mental health problems themselves (Runeson & Wilcox, 2021).

Critically, death by suicide is preventable through evidence-based interventions (WHO, 2018), with global suicide mortality declining by approximately one-third over the past three decades (Naghavi, 2019). However, these gains remain inequitably distributed, with low- to middle-income countries (LMICs) seeing comparatively small improvements, with some LMICs, such as Zimbabwe, Paraguay and Jamaica, showing increasing rates of suicide (Lovero et al., 2023). Today, LMICs account for 73% of all suicides (WHO, 2025). More broadly, there is growing evidence that humanitarian emergencies and fragile states, most of which unfold in LMICs (Al Omari et al., 2024), are associated with elevated risk of suicide (Jourdi and Kyrillos, 2022) and NSSI (Alem et al., 2021; Basu et al., 2022). Epidemiological studies highlight this vulnerability: refugee populations frequently exhibit higher rates of suicidal behaviour (IOM, 2017; Akinyemi et al., 2015) and NSSI (Gargiulo et al., 2020) than non-displaced groups, with conflict-affected societies bearing a similarly heavy toll (Al-Ahdal and Farahat, 2022; Sourander et al., 2024). Relatedly, survivors of natural disasters routinely report elevated rates of suicidal thoughts (Beaglehole et al., 2018) and increased likelihood of NSSI (Edwards et al., 2024). In addition to conflict and natural disasters, public health emergencies represent another form of crisis that can increase suicidal ideation (Cénat et al., 2020; Gunnell et al., 2020; Yan et al., 2023) and NSSI (Farooq et al., 2021) as strong predictors of suicide (Reeves et al., 2022; Moloney et al., 2024).

Several interrelated factors are thought to contribute to the increased risk of suicide and NSSI in humanitarian contexts (Jafari et al., 2020; IASC, 2022). In addition to individual-level factors (e.g., age, sex and prior trauma history; Knipe et al., 2022), humanitarian emergencies are characterised by disrupted or limited access to basic necessities, such as food, water, sanitation and safe shelter (IASC, 2007); forced displacement (Nguyen et al., 2023); increased rates of mental health disorders (Charlson et al., 2019); increased exposure to potentially traumatic events (Sabawoon et al., 2022); a lack of accessible care (Cogo et al., 2022); and the inability of governments to adequately promote suicide prevention (IASC, 2022). Simultaneously, disruptions to family cohesion and community networks diminish protective social supports (Jafari et al., 2020). Finally, humanitarian emergencies can exacerbate challenges arising from shortages of trained personnel, poor or unreliable referral pathways and the absence of practical tools for frontline workers to identify and assist high-risk individuals (UNHCR, 2023).

Despite this pressing need, suicide prevention has only recently begun to receive dedicated attention within humanitarian programming. Over the past decade, initiatives have included training frontline health workers on the World Health Organization's (WHO) Mental Health Gap Action Programme (mhGAP; Humayun et al., 2017; Keynejad et al., 2021), which includes content on suicide risk screening (WHO, 2015), campaigns to foster help-seeking behaviour (Schouler-Ocak, 2015) and deploying contact and safety planning interventions (Vijayakumar et al., 2017). Additionally, the Inter-Agency Standing Committee (IASC) – the World's 'longest-standing and highest-level humanitarian coordination forum' (IASC, 2025, para. 1) – recently developed its '*Addressing Suicide in Humanitarian Settings*' guidance note, which asserts that responding to suicide in emergency settings requires a multisectoral and collaborative approach (IASC, 2022).

Nonetheless, there are a few – yet heterogeneous – specific suicide prevention programmes. Previous evidence syntheses of suicide prevention interventions in humanitarian contexts have focused only on populations in displacement (Haroz et al., 2020) or have excluded grey literature and contexts of armed conflict (Reifels et al., 2024). Therefore, we set out to answer the following review question: Which suicide and self-harm prevention strategies have been implemented and evaluated in *all types of* humanitarian crises worldwide, and what is currently known about their effectiveness?

By synthesising this body of knowledge, we intend not only to highlight promising approaches but also to guide future empirical work and resource development – ultimately laying the groundwork for the development of robust, evidence-informed practical guidance to enhance the capacity of frontline humanitarian workers.

## Methods

We undertook a scoping review of the literature, conducted in accordance with the PRISMA extension for scoping revies (PRISMA-ScR) guidelines (Tricco et al., 2018). Please see Supplementary File 1 for a completed PRISMA-ScR checklist. No language or date restrictions were applied to the search, which was conducted in November 2024.

### Search strategy

A comprehensive search strategy was developed in collaboration with a subject librarian (GS) to identify relevant peer-reviewed literature across the following databases: CINAHL, Embase, MEDLINE, PsycINFO, Web of Science Core Collection and PTSDPubs. Search terminology spanned three domains: suicide/self-harm, humanitarian and fragile contexts, and intervention efficacy. Example search terms for each domain, respectively, included: suicid*, selfharm*, selfinjur*; disaster*, humanitarian, pandemic, fragile; intervention*, prevent*, effect*, outcome*. Supplementary File 2 contains our complete utilised search strings, formatted for MEDLINE (Ebsco).

Grey literature sources were identified through expert recommendations and searches of relevant organisational websites and grey literature databases (see Supplementary File 3).

Table 1 presents the criteria for inclusion in our review.

**Table 1.** Inclusion/exclusion criteria for record inclusion and data extraction

|  | Inclusion criteria | Exclusion criteria |
|---|---|---|
| Population | Human populations affected by disasters or emergencies (including all ages, nationalities, sexes and genders) | All other populations |
| Intervention | Studies examining interventions or programmes that had an aim of reducing suicide/self-harm-related outcomes and had some form of quantitative evaluation of intervention efficacy<br>This could include interventions aimed at increasing the capacity of caregivers to manage these outcomes among beneficiaries, so long as effectiveness was measured at the beneficiary level | Any other mental health intervention where its effects were not assessed for suicide, self-harm, NSSI and so on. Interventions described as national prevention strategies or suicide prevention policies |
| Context | Studies set in any humanitarian or fragile context, such as war, displacement, natural disasters (i.e., wildfires, earthquakes, etc.) or health emergencies (i.e., epidemic, pandemic, or other infectious disease outbreaks) | All other contexts |
| Outcome | Studies that included the following as either primary or secondary outcomes:<br>Suicide death<br>Suicide attempt<br>Suicidal thoughts/ideation<br>Suicide risk<br>Self-harm thoughts and/or behaviour<br>Non-suicidal self-injury<br>Or any other outcome directly related to suicide and/or self-harm | Accidental overdose<br>Medically assisted dying<br>Post-traumatic stress disorder (PTSD)<br>Any other mental health disorder |
| Study design | Quantitative (i.e., trials, pre-post studies, retrospective observational studies, etc.)<br>Mixed methods | All other study designs, including only qualitative |
| Types of articles | Empirical peer-reviewed studies, grey literature, dissertations/theses and conference abstracts (if they included sufficient details to describe the intervention and its effectiveness) | Reviews<br>Opinion/Commentaries<br>Descriptive studies (i.e., only reporting prevalence, demographics, etc., in relation to suicide/self-harm related outcomes)<br>Economic evaluation<br>Study protocols<br>Case reports |

### Screening and data extraction

All sources were uploaded to and deduplicated in Covidence (2024). Five authors were involved in the screening process (CZ, FV, FB, IK and RS). At both title/abstract and full text screening, each record was screened by two authors. Any discrepancies between the first two screeners were resolved by discussion and involving a third screener, if necessary. Data were extracted for the following domains: study characteristics, intervention details, suicide/self-harm-related outcomes, and efficacy of the intervention (see Supplementary File 4). Data extraction was first completed independently by two authors (CZ and RS), who then met to discuss any discrepancies.

### Quality assessment

The Mixed Methods Appraisal Tool (MMAT; Hong et al., 2018) was applied to assess the quality of each included study, chosen for its capacity to appraise multiple empirical study designs. For each study design category, reviewers respond 'yes' (=1), 'no' (=0) or 'cannot tell' (=0) to five questions related to methodological rigour. Each study is thus assigned a quality score, ranging from 0 (0%) to 5 (100%). Two authors (CZ and RS) first independently assessed each study and subsequently discussed any discrepancies. As per the MMAT, studies were not excluded based on methodological quality, but those deemed of the highest quality were prioritised in our reporting. Table 2 provides a 0–100% quality score for each study, in accordance with the MMAT.

### Results

Our search of the peer-reviewed literature identified a total of $n = 9,824$ records, including 3,615 duplicates. Of the remaining 6,209 screened at the title/abstract phase, 104 were included for full text screening. Twenty-three studies were included for data extraction.

Our grey literature search identified 712 relevant records (Supplementary File 2). These were reviewed by one author, and 29 records were included for full-text review. None of these, however, met our full inclusion criteria. Figure 1 summarises the screening process.

### Study characteristics

Characteristics of the 23 included studies and interventions are presented in Table 2. All studies were published between 2003 and 2024.

### Study design

The largest proportion of studies employed a non-randomised experimental approach ($n = 9$, 39.1%; Abdulah and Abdulla, 2020; Agyapong et al., 2021; Agyapong et al., 2023; Anichini et al., 2020; Dias et al., 2023; Kim et al., 2020; Obuobi-Donkor et al., 2024; Vijayakumar and Kumar, 2008; Won et al., 2023), followed by randomised controlled trials ($n = 6$; Ertl et al., 2011; Devassy et al., 2021; Persich et al., 2021; Bryant et al., 2023; Dominguez-Rodriguez et al., 2023, 2024). Four studies employed

**Table 2.** Overview of included studies and their evaluated interventions

| Authors (year) | Study location | Study design | Participant description | Intervention setting | Intervention description | Intervention training | Suicide-related outcomes (mode of assessment) | Main findings | Quality of Study* |
|---|---|---|---|---|---|---|---|---|---|
| Abdulah and Abdulla (2020) | Iraq | Non-randomised experimental (pre-post without a control group) | Kurdish Yezidi women and girls, internally displaced persons (IDPs) who had survived captivity or fled following an attack by the Islamic State of Iraq (ISIS) $N = 14$ Age (range): 10–29 years | IDP camp | **Content**: Creative art intervention, with a camp exhibition of participant art at the end **Duration**: 2 months **Format**: Group **Mode of delivery:** In-person **Provider**: Specialist | N/A | Suicidal ideation (BSS[1]) | **After 2 months:** Significant reduction in total scores of suicidal ideation ($M_{pre}$ = 16.71, [SD = 9.34] vs. $M_{post}$ = 6.50, [SD = 1.02], $p$ = 0.002) | 60% |
| Agyapong et al. (2021) | Alberta, Canada | Non-randomised experimental (naturalistic controlled trial) | Adult subscribers to Text4Hope messaging service $N = 2,767$ (2,011 in intervention group [IG], 756 in control group [CG]) Age (range): 18+ years | COVID–19 | **Content:** Self-subscription text-based service whereby subscribers receive daily supportive SMS text messages that have been written by mental health professionals using a cognitive behavioural framework **Duration:** Not specified **Format:** Individual **Mode of delivery:** remote **Provider:** self-subscription | N/A | Thoughts of self-harm or death wish (PHQ–9[2], item 9) | **After 6 weeks:** Significantly lower prevalence of thoughts of self-harm or death wish in IG compared to CG (16.9% vs. 26.6%, $p$ < 0.001), with a small negative effect size (Phi = −0.106) Significant reduction in the likelihood of having thoughts of self-harm or death wish among IG (OR = 0.59, 95% CI = 0.45–0.77, $p$ < 0.001) | 60% |
| Agyapong et al. (2023) | Alberta, Canada | Non-randomised experimental (naturalistic controlled trial) | | | **Content:** Self-subscription text-based service whereby subscribers receive daily supportive SMS text messages that have been written by mental health professionals using a cognitive behavioural framework **Duration:** Not specified **Format:** Individual **Mode of delivery:** Remote **Provider:** Self-subscription | N/A | Thoughts of self-harm or death wish (PHQ–9[2], item 9) | **After 6 weeks:** Significantly lower prevalence of thoughts of self-harm or death wish in IG compared to the CG (40.3% vs. 59.8%, $p$ = 0.01), with a small effect size (Phi/Cramer's $V$ = 0.19) Significant reduction in the likelihood of having thoughts of self-harm or death wish among IG (OR = 0.42, 95% CI = 0.21–0.92, $p$ = 0.01) | 80% |
| Anichini et al. (2020) | Italy | Non-randomised experimental (pre-post | Children and adolescents already in treatment for severe and complex psychopathology at a | COVID–19 | **Content:** Multidisciplinary intervention, including psychiatric services, art therapy, psychotherapy | N/A | Suicidal ideation and behaviour (clinical interviews) Non-suicidal self- | **After 90 days:** Non-significant reduction in prevalence of suicidal ideation (35.4% vs. 20.8%, $p$ = 0.092) and | 40% |

(*Continued*)

| Authors (year) | Study location | Study design | Participant description | Intervention setting | Intervention description | Intervention training | Suicide-related outcomes (mode of assessment) | Main findings | Quality of Study* |
|---|---|---|---|---|---|---|---|---|---|
| | | without a control group) | psychiatric-therapeutic day hospital $N = 48$ Age (range): 9–19 years | | and educational interventions **Duration:** 90 days **Format:** Group and individual **Mode of delivery:** Remote **Provider:** Specialist | | injury (NSSI; clinical interviews) | NSSI (22.9% vs. 12.5%, $p = 0.063$) | |
| Bryant et al. (2023) | Australia | Randomised controlled trial (RCT) | Adults who screened positive for COVID–19-related psychological distress $N = 174$ (87 in IG; 87 in CG) Age (range): 18+ years | | **Content:** Positive affect training **Duration:** 6 weeks **Format:** Group **Mode of delivery:** Remote **Provider:** Specialist | N/A | Suicidal ideation (SIDAS[3]) | Significantly greater reductions in suicidal ideation compared to the CG after 7 weeks (mean difference = 4.3, $p = 0.03$) and 3 months (mean difference = 5.0, $p = 0.006$) Mean differences at both follow-ups showed a moderate effect size (0.4, 95% CI = 0.10–0.8) | 80% |
| Devassy et al. (2021) | India | RCT | Upskilled youth from Deen Dayal Upadhyaya Grameen Kaushalya Yojan (DDUGKY) centres, which aim to provide quality training and work placements to underserved, poor rural youth across India $N = 439$ (251 in IG; 188 in CG) Age (mean [SD]): 25.1 (5.7) years | COVID–19 | **Content:** Befriending intervention focused on proactive engagement/crisis intervention, problem-solving, supportive therapy and linkage with community resources **Duration:** 1 month **Format:** Individual **Mode of delivery:** Remote **Provider:** Lay individuals | **Recipients**: REaCH intervention team (DDUGKY staff members with at least one year of experience) **Content:** Training on the content and process of intervention. Recipients were also provided with an intervention manual, a video of the training material, audio clips of model interviews and a module on frequently asked questions **Duration:** One day (6 h) **Mode of delivery:** online **Provider:** not specified | Suicidality (not specified) | **After 1 month:** Non-significant reduction in mean score of suicidality among participants in the IG ($M_{pre} = 0.25$ [SD = 0.6], $M_{post} = 0.24$ [SD = 0.6], $p = 0.55$) Lower, but non-significant, likelihood of reporting suicidality among participants in the IG compared to CG (OR = 0.80, 95% CI = 0.59–1.08, $p = 0.156$) | 80% |
| Dias et al. (2023) | Alberta, Canada | Non-randomised experimental (naturalistic controlled trial) | Adult female subscribers to Text4Hope messaging service $N = 2,330$ (1,763 in IG; 567 in CG) Age (range): 18+ years | COVID–19 | **Content:** Self-subscription text-based service whereby subscribers receive daily supportive SMS text messages that have been written by mental health professionals using a cognitive | N/A | Thoughts of self-harm or death wish (PHQ–9[2], item 9) | **After 6 weeks:** Significantly lower prevalence of thoughts of self-harm or death wish in IG compared to the CG (15.7%% vs. 26.1%, $p < 0.001$), with a small effect size (Phi/Cramer's $V = 0.116$) | 60% |

**Table 2.** (*Continued*)

| Authors (year) | Study location | Study design | Participant description | Intervention setting | Intervention description | Intervention training | Suicide-related outcomes (mode of assessment) | Main findings | Quality of Study* |
|---|---|---|---|---|---|---|---|---|---|
| | | | | | behavioural framework **Duration:** Not specified **Format:** Individual **Mode of delivery:** Remote **Provider:** Self-subscription | | | Significant reduction in the likelihood of having thoughts of self-harm or death wish among IG (OR = 0.55, 95% CI = 0.41 to 0.73, $p < 0.001$) | |
| Dominguez-Rodriguez et al. (2023) | Latin America (96.5% of participants were from Mexico) | RCT | Spanish-speaking adults who had lost a loved one in the 6 months before participation and reported symptoms of depression, anxiety or stress $N$ = 114 (45 in IG; 69 in CG) Age (range): 21–62 years | COVID–19 | **Content:** Multimodal, combining techniques of positive psychology, cognitive behavioural therapy (CBT), behavioural activation (BA) therapy and mindfulness **Duration:** 36 days **Format:** Individual **Mode of delivery:** Remote **Provider:** Self-administered | N/A | Suicide risk (PSRS[4]) | Participants in the IG experienced a statistically significant reduction in median scores of suicide (medianpre = 3 [IQR = 1–6], medianpost = 2 [IQR = 1–3], $p = 0.004$), which demonstrated a medium effect size (Hedge's $g$ = 0.5). This difference was sustained at the 3-month follow-up, with participants who completed follow-up reporting a statistically significant ($p = 0.001$) reduction in median suicide risk scores from 4 (IQR = 3–6; pretreatment) to 1 (IQR = 1–3; 3-month follow-up). Participants in the CG reported a statistically significant increase in median scores of suicide risk immediately after the intervention period (medianpre = 3 [IQR = 2–5], medianpost = 4 [IQR = 2–5], $p = 0.004$) | 20% |
| Dominguez-Rodriguez et al. (2024) | Mexico | RCT | Mexican adults with access to a technological device $N$ = 36 (31 in IG; 5 in CG) Age (range): 18–64 years | COVID–19 | **Content:** Multi-modal intervention based on positive psychology, CBT and BA therapy The CG received the intervention as described above, while the IG received the intervention as described above with | N/A | Suicidal ideation (BSS[1]) | **Immediately following intervention:** Non-significant reduction in the proportion of participants in the IG who met the threshold for suicidal ideation ($n = 7$, 23.3% vs. $n = 6$, 20.0%, $p = 0.56$) No participants in the | 20% |

(*Continued*)

Cambridge Prisms: Global Mental Health

| Authors (year) | Study location | Study design | Participant description | Intervention setting | Intervention description | Intervention training | Suicide-related outcomes (mode of assessment) | Main findings | Quality of Study* |
|---|---|---|---|---|---|---|---|---|---|
| | | | | | the addition of a chat support feature. Chat support was provided by therapists in training and included offering support on the web-based platform and intervention content **Duration**: Self-paced **Format**: Individual **Mode of delivery:** Remote **Provider:** Self-administered | | | CG met the threshold for suicidal ideation at either baseline or follow-up | |
| Ertl et al. (2011) | Uganda | RCT | Former Ugandan child soldiers with post-traumatic stress disorder (PTSD) residing in IDP camps $N = 85$ (57 in IGs; 28 in CG) Age (range): 12–25 years | IDP camp | **Content:** Narrative exposure therapy ($n = 29$) and academic catch-up with counselling ($n = 28$) **Duration:** 3 weeks **Format:** Individual **Mode of delivery:** In-person **Provider:** Lay individuals | **Recipients:** Local lay counsellors **Content:** Training was conducted using an adapted field version of the narrative exposure therapy training manual **Duration: N**ot specified **Mode of delivery: N**ot specified **Provider:** Not specified | Suicide risk (Module C of the MINI[5]) | **After 12 months**: Participants in both the narrative exposure and academic catch-up IGs reported non-significant reductions in suicide risk scores, with mean changes of $-7.36$ (SE = 2.89) and $-4.26$ (SE = 1.78), respectively. These reductions indicated moderate effect sizes (Cohen's $d$ of 0.69 and 0.55, respectively) | 80% |
| Gliske et al. (2022) | United States of America (USA) | Retrospective observational cohort study with a pre-post analysis | Youth and young adults with high-acuity and co-occurring mental and behavioural health needs $N = 495$ Age (range): 11–35 years | COVID–19 | **Content:** Intensive outpatient programme (IOP) employing a multi-modal therapeutic approach, including family therapy, art-based therapy and mindfulness. Dialectical behavioural therapy (DBT) is provided for individuals with high suicide risk **Duration:** Not specified **Format:** Group and individual **Mode of delivery:** Remote **Provider:** Not specified | N/A | Suicidality (ASQ[6]) Frequency of NSSI (Criterion A of the ABASI[7]) | **After 2 weeks:** Significant reduction in the number of participants screening positive for suicide risk ($n = 330$ vs. $n = 115$, $p < 0.001$) Significant reduction in the number of participants screening positive for active suicidal ideation ($n = 201$ vs. $n = 92$, $p < 0.001$) Significant reduction in the number of participants meeting criteria for NSSI ($n = 205$ vs. $n = 119$, $p < 0.001$) Significant reduction in mean scores of NSSI | 80% |

**Table 2.** (*Continued*)

| Authors (year) | Study location | Study design | Participant description | Intervention setting | Intervention description | Intervention training | Suicide-related outcomes (mode of assessment) | Main findings | Quality of Study* |
|---|---|---|---|---|---|---|---|---|---|
| | | | | | | | | frequency (12.09 vs. 6.08, $t_{429}$ = 10.41, $p < 0.001$) | |
| Gujral et al. (2022) | USA | Retrospective observational cohort study with a pre-post analysis | Rural veterans in the United States with at least one visit to the Department of Veterans Affairs (VA) hospitals with at least one VA visit in 2019 $N$ = 471,791 (13,180 in IG; 458,611 in CG) Age (mean [SD]): 61.2 (13.4) years | COVID–19 | **Content:** Provision of video-enabled tablets **Duration:** Not specified **Format:** Individual **Mode of delivery:** Remote **Provider:** Not specified | N/A | Likelihood of suicide-related emergency department (ED) visits (Program Evaluation Resource Center of the VA [PERC]) Number of suicide behaviour and overdose reports (SBORs; PERC) | **After 10 months:** Significant decrease of 36% in the likelihood of a suicide-related ED visit (proportion change: −0.0017; 95% CI = −0.0023 to −0.0013) Significant decrease of 22% in number of SBORs (monthly coefficient: −0.0011, 95% CI = −0.0016 to −0.0005) | 80% |
| Kelly et al. (2003) | United Kingdom (Northern Ireland) | Retrospective observational correlational study | Residents of Northern Ireland between the years 1989 and 1999 Age (range): All ages (in years) were included. The sample was divided into those <30 years of age and individuals ≥30 years | Armed conflict | **Content:** Rate of prescription of anti-depressant medications between 1989 and 1999 **Duration:** Not applicable (n/a) **Format:** n/a **Mode of delivery:** n/a **Provider:** n/a | N/A | Recorded cases of suicide and undetermined deaths between 1989 and 1999 (General Register Office, Belfast, Northern Ireland) | There was no significant association between suicide and antidepressant prescribing for people <30 years of age within the sample For the older age group (≥30 years), there was a significant inverse relationship between antidepressant prescribing and suicide ($t$ = − 2.90, $p$ = 0.02) | 100% |
| Kim et al. (2020) | South Korea | Non-randomised experimental (pre-post without a control group) | Individuals hospitalised for COVID–19 $N$ = 33 Age (mean [SD]): 45 (18.34) years | COVID–19 | **Content:** Psychological intervention programme and/or pharmacotherapy, depending on individual need **Duration:** 2 weeks **Format:** Individual **Mode of delivery:** Remote **Provider:** Specialist | N/A | Suicidal ideation (BDI[8], item 9) | **After 2 weeks:** Non-significant reduction in the number of participants screening positive for suicidal ideation ($n$ = 3 vs. $n$ = 1, $p$ = 0.396). | 40% |
| Landrum et al. (2023) | Malawi | Implementation science trial | Individuals in care for diabetes or hypertension management at participating non-communicable disease clinics who also had elevated depression symptoms | COVID–19 | **Content:** A telephone-delivered suicide risk assessment (SRA) protocol to assess and reduce suicidal ideation, designed to follow a positive suicide risk assessment. The SRA involves assessment of | **Recipients**: Research assistants on the study team **Content:** Training on content and implementation of the telephone-based SRA, as well as practice of its implementation in | Suicidal ideation (PHQ–9[2], item 9; $n$ = 13; OR through other not specified methods; $n$ = 2) | There was a 100% resolution rate of suicidal ideation among the $n$ = 15 participants who screened positive for suicidal ideation and received the SRA intervention. Statistical significance was not | 60% |

*Cambridge Prisms: Global Mental Health*

| Authors (year) | Study location | Study design | Participant description | Intervention setting | Intervention description | Intervention training | Suicide-related outcomes (mode of assessment) | Main findings | Quality of Study[*] |
|---|---|---|---|---|---|---|---|---|---|
| | | | $N = 738$ (15 of which received intervention by screening positive for suicidal risk) Age (range): 18–65 years | | suicidal ideation, creation of a safety and follow-up plan, and additional follow-up actions depending on the level/type of the participant's suicidal ideation (i.e., passive vs. active) **Duration:** Depends on level/type of suicidal ideation. Average duration among participants was 13.96 days (range 0–35 days) **Format:** Individual **Mode of delivery:** Remote **Provider:** Lay individuals | simulated scenarios. A follow-up meeting occurred a week after training to discuss any questions around the implementation of the protocol **Duration:** Two 1-h trainings **Mode of delivery:** Online **Provider:** Clinical and research study team members | | reported | |
| Obuobi-Donkor et al. (2024) | Alberta and Novia Scotia, Canada | Naturalistic controlled trial | Adult subscribers to Text4Hope messaging service $N = 226$ (174 in IG; 52 in CG) Age (range): 18+ years | Natural disaster (wildfire) | **Content:** Self-subscription text-based service whereby subscribers receive daily supportive SMS text messages that have been written by mental health professionals using a cognitive behavioural framework **Duration:** Not specified **Format:** Individual **Mode of delivery:** Remote **Provider:** Self-subscription | N/A | Thoughts of self-harm or death wish (PHQ–9[2], item 9) | **After 6 weeks:** Significantly lower ($t = 3.85$, $p < 0.001$) mean score of thoughts of self-harm or death wish in IG ($M = 0.27$ [SD = 0.6]) compared to CG ($M = 0.60$ [SD = 1.0]). This difference indicated a moderate effect size (Hedge's $g = 0.57$) | 60% |
| Persich et al. (2021) | USA | RCT | Adults from a university setting and the surrounding general population $N = 89$ (52 in IG; 37 in CG) Age (mean [SD]) = 23.5 (5.59) years | COVID–19 | **Content:** Emotional intelligence training programme **Duration:** Self-paced (takes about 9–11 h to complete) **Format:** Individual **Mode of delivery:** Remote **Provider:** Self-administered | N/A | Suicidal ideation (BDI[8], item 9) | **After 6 months:** Significantly lower mean score of suicidal ideation among the IG (0.08) compared to CG (0.24; F[2, 174] = 3.79, $p = 0.024$) | 20% |

**Table 2.** (Continued)

| Authors (year) | Study location | Study design | Participant description | Intervention setting | Intervention description | Intervention training | Suicide-related outcomes (mode of assessment) | Main findings | Quality of Study* |
|---|---|---|---|---|---|---|---|---|---|
| Puspitasari et al. (2021) | USA | Retrospective observational cohort study with a pre-post analysis | Individuals with transdiagnostic mental health conditions who were at risk of psychiatric hospitalisation $N = 76$ Age (range; mean [SD]): 18–73 years; 36.6 (13.4) years | COVID–19 | **Content:** IOP programme with psychotherapy content focused on CBT, DBT and BA therapy and process-based therapy **Duration:** 3 weeks **Format:** Group and individual **Mode of delivery:** Remote **Provider:** Specialist | N/A | Suicide risk; wish to die; wish to live (SSF[9]) | **After 3 weeks:** Significant reductions in mean scores of suicide risk ($M_{pre} = 1.53$ [SD = 0.82] vs. $M_{post} = 1.24$ [SD = 0.58], $p = 0.02$), and wish to die ($M_{pre}$1.67, $M_{post} = 0.74$, $p = 0.01$). These reductions indicated a small-moderate effect size (Cohen $d = 0.41$ and 0.52, respectively) Significant increase in mean score of wish to live ($M_{pre} = 6.26$, $M_{post} = 6.95$, $p < 0.001$). This increase indicated a small effect size (Cohen $d = 0.39$) | 80% |
| Ramaiya et al. (2022) | Nepal | Mixed methods | Secondary school students $N = 102$ (42 in IG, 60 in CG) Age (range): 13–17 years | Natural disaster (earthquake) | **Content:** DBT-informed, emotion-focused skills training programme **Duration:** 4 weeks **Format:** Group **Mode of delivery:** In-person **Provider:** Lay individuals | N/A | Suicidal ideation (4-item scale with 'yes' and 'no' response options; assessed by a member of the research team with prior training in Nepali suicide prevention) | **After four weeks:** Reduction of 23% in the prevalence of suicidal ideation among participants in IG ($n = 13$ vs. $n = 10$). No difference was found in the prevalence of suicidal ideation in CG from baseline ($n = 15$ vs. $n = 15$). Statistical significance could not be determined due to the small base rate of suicidal ideation by condition (IG vs. CG) | 60% |
| Stevens et al. (2022) | United Kingdom | Mixed methods | Children and young people accessing the Kooth platform for the first time $N = 302$ Age (range): 13–21 years | COVID–19 | **Content:** Online platform that provides users access to professional counselling, peer support and well-being-related activities **Duration:** Not specified **Format:** Not specified **Mode of delivery:** Remote **Provider:** Self-administered and specialist | N/A | Suicidal ideation (SIDAS[3]) Self-harm (assessed via the question, 'In the past month, have you ever deliberately hurt yourself or done anything that might have harmed you or even killed you?' | **After 1 month: Among the $n = 133$ participants who only used the peer support feature:** Significant reduction in mean scores of suicidal ideation ($M_{pre} = 15.1$, $M_{post} = 12.8$, $p = 0.005$) Significant reduction in proportion reporting self-harm ($M$pre = 0.4, $M$post = 0.3, $p = 0.033$) **Among the $n = 151$** | 40% |

(Continued)

*Cambridge Prisms: Global Mental Health*

| Authors (year) | Study location | Study design | Participant description | Intervention setting | Intervention description | Intervention training | Suicide-related outcomes (mode of assessment) | Main findings | Quality of Study* |
|---|---|---|---|---|---|---|---|---|---|
| | | | | | | | | **participants who used both the peer support and professional counselling feature:** Significant reduction in mean scores of suicidal ideation ($M_{pre}$ = 15.7, $M_{post}$ = 13.8, $p$ = 0.011) Significant reduction in proportion reporting self-harm ($M_{pre}$ = 0.4, $M_{post}$ = 0.3, $p$ = 0.011) **Among all ($n$ = 302) participants:** Significant reduction in mean scores of suicidal ideation ($M_{pre}$ = 16.5, $M_{post}$ = 15.1, $p$ = 0.007) Significant reduction in proportion reporting self-harm ($M_{pre}$ = 0.5, $M_{post}$ = 0.4, $p$ = 0.001) | |
| Vijayakumar and Kumar (2008) | India | Non-randomised experimental study | Non-migrant adults (aged 18 years or more) from two different coastal regions (serving as the IG and CG) in India who had lost at least one family member during the 2004 Asian tsunami $N$ = 102 (45 in IG, 57 in CG) Age (mean [SD]): 38.2 (14.0) | Natural disaster (tsunami) | **Content:** Befriending intervention involving the provision of regular human contact and emotional support, offering availability, unconditional acceptance, total confidentiality and empathy to the recipient **Duration:** Not specified **Format:** Individual **Mode of delivery:** In-person **Provider:** Lay individuals | **Recipients:** Volunteers with at least 4 years of experience from a local suicide prevention organisation **Content:** Training on how to provide emotional assistance to bereaved individuals **Duration:** 8 h **Mode of delivery:** Not specified **Provider:** Not specified | Suicide attempt (not specified) | Over the course of 12 months following delivery of the intervention, significantly fewer individuals in the IG attempted suicide compared to those in the CG ($n$ = 0 vs. $n$ = 3, $p$ = 0.02) | 40% |
| Vijayakumar et al. (2017) | India | Mixed methods | Sri Lankan refugees residing in two distinct refugee camps (serving as the IG and CG) in Southern India $N$ = 1,303 (639 in IG, 664 in CG) Age (range; mean [SD; IG, CG]): 18+ years; 41.58 (15.0), 39.10 (15.0) years | Refugee camp | **Content:** Regular contact (focused on emotional support) and safety planning **Duration:** Not specified **Format:** Individual **Mode of delivery:** In-person **Provider:** Lay individuals | **Recipients:** Women residing in the refugee camps. Training recipients were selected based on interviews assessing their willingness to deliver the intervention, as well as their ability to empathise and maintain confidentiality **Content:** Information | Death by suicide (rates per 100,000 population) Suicide attempt (rates per 100,000 population) Suicidal ideation (BSS[1]) | **After 15 months:** Reduction in rate per 100,000 of suicide attempt (671/100,000 vs. 371/100,000) AND suicide attempt + death by suicide combined (964/100,000 vs. 445/100,000) in IG Significant difference in the change in rate per 100,000 of suicide | 40% |

**Table 2.** (*Continued*)

| Authors (year) | Study location | Study design | Participant description | Intervention setting | Intervention description | Intervention training | Suicide-related outcomes (mode of assessment) | Main findings | Quality of Study* |
|---|---|---|---|---|---|---|---|---|---|
| | | | | | on loss and grief, depression and suicide, as well as training in communication skills and empathetic offering of emotional support. Recipients were also trained in obtaining informed consent and in completing the safety planning cards for intervention participants **Duration:** 20 h **Mode of delivery:** Not specified **Provider:** Not specified | | | attempt between IG and CG (296/100,000, 95% CI = 6.7–587, $p = 0.05$) Significant difference in the change in rate per 100,000 of suicide attempt + death by suicide between IG and CG (519/100,000, $p = 0.01$) | |
| Won et al. (2023) | South Korea | Non-randomised experimental (pre-post without a control group) | Individuals in the COVID–19 inpatient ward of a hospital $N = 32$ Age (mean [SD]): 50.56 (17.42) years | COVID–19 | **Content:** Psychiatric consultation programme involving education on COVID–19, stress management and relaxation therapy. For participants who indicated ideation of suicide and/or NSSI, psychiatrists delivered in-person emotional support, assistance in meeting practical needs and future disposition planning **Duration:** Not specified **Format:** Individual **Mode of delivery:** Remote/in-person **Provider:** Specialist | N/A | Suicide risk (P4 Suicidality Screener[10]) | Non-significant reductions in mean score of suicide risk following intervention ($M_{pre} = 0.22$ [SD = 0.42], $M_{post} = 0.00$ [SD = 0.00], $p = 0.083$) | 60% |

*M*, mean; SD, standard deviation; OR, odds ratio; CI, confidence interval; SE, standard error; IQR, interquartile range.

*Note*: [1]Beck Scale for Suicide Ideation (Beck et al., 1988); [2]Patient Health Questionnaire-9 (Kroenke et al., 2001); [3]Suicidal Ideation Attributes Scale (van Spijker et al., 2014); [4]Plutchik Suicide Risk Scale (Plutchik and Van Praag, 1994); [5]Mini International Neuropsychiatric Interview – English Version 5.0.0 (Sheehan et al., 1998); [6]Ask Suicide-Screening Questions (Horowitz et al., 2012); [7]Alexian Brothers Assessment of Self-Injury (Washburn et al., 2015); [8]Beck Depression Inventory (Beck et al., 1961); [9]Suicide Status Form (Conrad et al., 2010); [10]P4 Suicidality Screener (Dube et al., 2010).
*According to the Mixed Methods Appraisal Tool (MMAT; Hong et al., 2018), which scores a study out of five criteria with quality scores ranging from 0 to 100%.

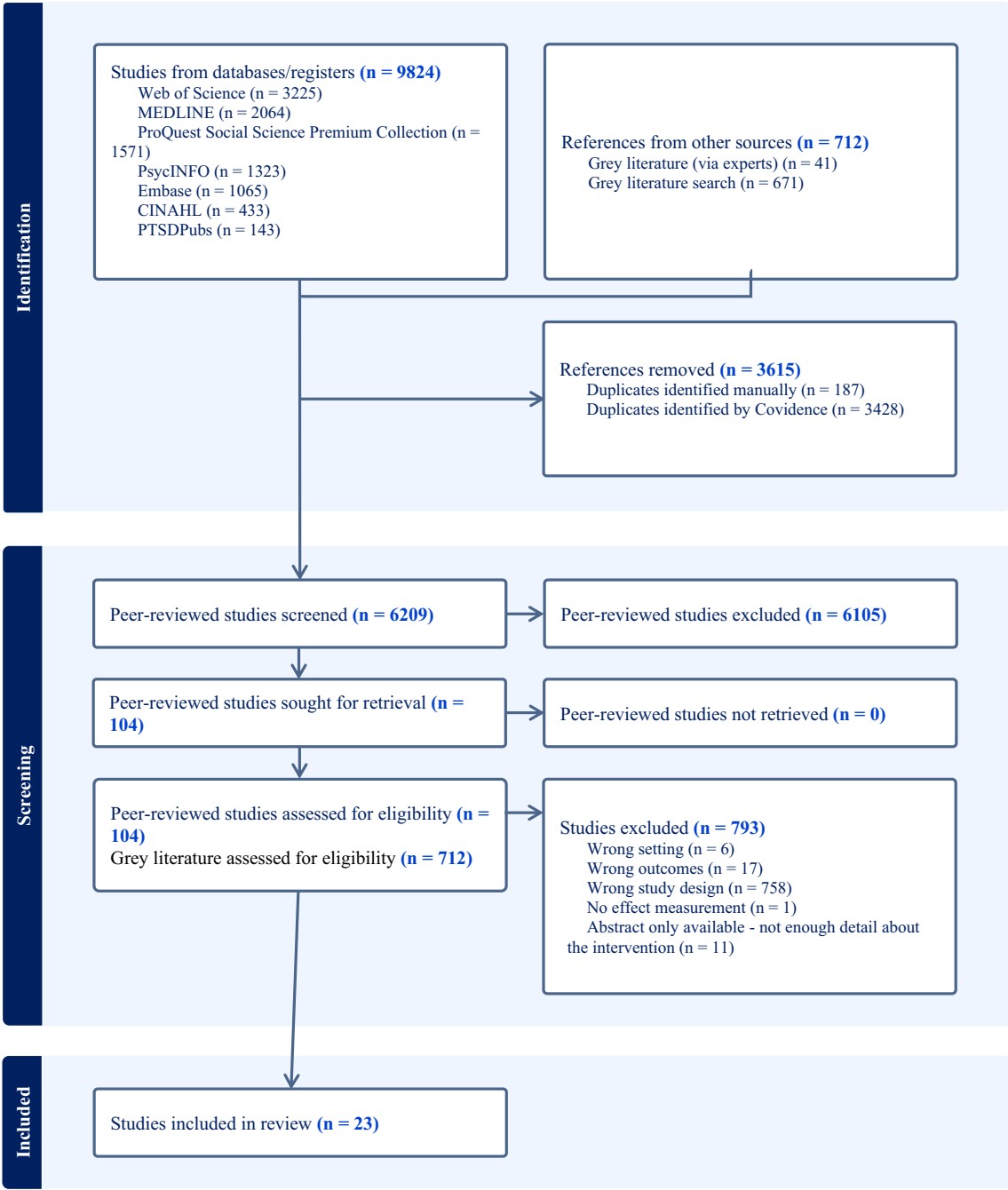

**Figure 1.** PRISMA 2020 flow diagram generated through Covidence (2024).

a retrospective observational design (Kelly et al., 2003; Puspita-sari et al., 2021; Gliske et al., 2022; Gujral et al., 2022). Three studies used mixed methods (all of which employed non-randomised experimental quantitative methods; Ramaiya et al., 2022; Stevens et al., 2022; Vijayakumar et al., 2017) and one study used implementation science (Landrum et al., 2023).

### Sample descriptions

Most studies (*n* = 14, 60.7%) focused on adult (aged ≥18 years) populations, while eight studies involved children, adolescents and young adults. The remaining study involved publicly accessible

data from all individuals residing in Northern Ireland between the years of 1989–1999 (Kelly et al., 2003).

### Intervention details

#### Intervention context

Of the 23 studies in our analysis, the majority (*n* = 16, 69.6%) examined interventions implemented or evaluated during the coronavirus disease 2019 pandemic (COVID-19). Most occurred in the United States (Persich et al., 2021, Puspitasari et al., 2021; Gliske et al., 2022; Gujral et al., 2022), Canada (Agyapong et al., 2021, 2023; Dias et al., 2023) or other high-income countries (Anichini et al.,

2020; Kim et al., 2020; Stevens et al., 2022; Bryant et al., 2023; Won et al., 2023). The remaining four interventions implemented during COVID-19 occurred in India (Devassy et al., 2021), Mexico (Dominguez-Rodriguez et al., 2023, 2024) and Malawi (Landrum et al., 2023). The second most prevalent humanitarian contexts were natural disasters – occurring in Canada (Obuobi-Donkor et al., 2024), Nepal (Ramaiya et al., 2022) and India (Vijayakumar and Kumar, 2008). The remaining intervention contexts were internally displaced person camps located in Iraqi Kurdistan (Abdulah and Abdulla, 2020) and Northern Uganda (Ertl et al., 2011), refugee camps in India (Vijayakumar et al., 2017) and the armed conflict in Northern Ireland (Kelly et al., 2003).

### Description of interventions

The largest proportion ($n = 8$, 34.8%) of studies examined interventions that either entirely or predominantly involved a psychotherapeutic approach, seven of which were delivered remotely. Four studies evaluated *Text4Hope*, a self-subscription, automated text messaging service that sends users daily messages informed by a cognitive behavioural therapy (CBT) framework (Agyapong et al., 2021, 2023; Dias et al., 2023; Obuobi-Donkor et al., 2024). Two studies investigated similar modularised, self-administered online platforms housing content based on CBT, behavioural activation (BA) therapy and positive psychology – with additional incorporation of mindfulness practices (Dominguez-Rodriguez et al., 2023) or chat support from therapists-in-training (Dominguez-Rodriguez et al., 2024). The remote, mental health specialist-led (hereafter referred to as 'specialist-led') intensive outpatient programme (IOP) evaluated by Puspitasari et al. (2021) involved a similar multitude of psychotherapies, where high-risk participants engaged in group-based BA therapy, dialectical behavioural therapy (DBT), and occupational therapy (OT). Finally, Ertl et al. (2011) investigated the in-person delivery of narrative exposure therapy and academic catch-up with elements of supportive counselling, administered by trained local 'lay' (i.e., non-specialist) counsellors.

Four studies examined interventions utilising multiple therapeutic models, each delivered remotely. The IOP evaluated by Gliske et al. (2022) involved primarily group-based therapies of both a psychotherapeutic and experiential (i.e., mindfulness and creative arts) nature, with individuals at high risk of suicide participating in DBT groups. Anichini et al. (2020) investigated a specialist-led intervention that offered a wide range of services, including art therapy workshops, group and individual psychotherapy and neuropsychiatric consultations. Kim et al. (2020) evaluated a specialist-led intervention featuring psychoeducation on COVID-19, CBT techniques, and psychotropic medication, when required. Finally, Stevens et al. (2022) evaluated *Kooth*, an online platform with self-administered well-being activities, a moderated peer support platform, and access to professional counselling.

Five studies evaluated interventions that provided direct crisis intervention of both a therapeutic and/or practical nature. Three of these were administered remotely. Devassy et al. (2021) assessed a telephone-based befriending intervention, administered by trained lay individuals, which focused on proactive engagement and crisis intervention, problem-solving oriented supportive therapy, and linking in with community resources. An additional remote intervention was a telephone-based suicide risk assessment protocol and subsequent safety planning delivered by trained lay individuals (Landrum et al., 2023). Won et al. (2023) examined a telephone-delivered, specialist-led psychiatric consultation programme that included education on COVID-19, stress management, and

relaxation therapy. For high-risk individuals, intervention activities shifted to in-person provision of emotional support, assistance in meeting practical needs, and future disposition planning. Two interventions were delivered in-person: Vijayakumar et al. (2017) evaluated *Contact and Safety Planning* (*CASP*), involving the provision of emotional support and safety planning by trained lay individuals, while Vijayakumar and Kumar (2008) evaluated a lay-delivered befriending intervention that centred on regular contact and emotional support for recently bereaved individuals.

Three studies evaluated skills-based training programmes, two of which were remote. Persich et al. (2021) investigated a brief, self-administered online emotional intelligence (EI) training, with Bryant et al. (2023) investigating a specialist-led group-based positive affect training. Ramaiya et al. (2022) evaluated a DBT-informed, emotion-focused training programme delivered to groups in-person by trained lay individuals.

Additional interventions included the in-person delivery of group-based creative arts therapy – administered by a creative arts specialist (Abdulah and Abdulla, 2020), the provision of video-enabled tablets (Gujral et al., 2022) and antidepressant medication (Kelly et al., 2003).

Table 2 provides more information around the content/duration of each intervention, as well as the training content for the five lay-delivered interventions.

### Outcomes and modes of assessment

Most studies focused on individual-level suicide/NSSI-related outcomes, employing a variety of assessment methods. The most frequent method was through validated measurement tools ($n = 17$, 73.9%). Of these, Item 9 of the Patient Health Questionnaire-9 (Kroenke et al., 2001), a measure of suicidal ideation and/or thoughts of NSSI, was used most frequently (Agyapong et al., 2021, 2023; Dias et al., 2023; Landrum et al., 2023; Obuobi-Donkor et al., 2024). In addition to using the Beck Scale for Suicidal Ideation (Beck et al., 1988) to assess individual suicidal ideation, Vijayakumar et al. (2017) also assessed rates of death by suicide and suicide attempt per 100,000 individuals in two refugee camps. Table 2 presents additional validated measurement tools used to assess suicide/NSSI-related outcomes.

One study assessed suicidal ideation/behaviour and NSSI through clinical interviews (Anichini et al., 2020) and another assessed suicidal ideation through a four-item scale developed by the authors (Ramaiya et al., 2022). Two studies drew from public records, one of which assessed the likelihood of a suicide-related emergency department visit and the number of suicide behaviour and overdose reports (SBORs) among US rural veterans (Gujral et al., 2022), while the other used the recorded cases of suicide and undetermined deaths across 10 years in Northern Ireland (Kelly et al., 2003).

Two studies did not report their mode of assessment for their suicide-related outcome of interest (Vijayakumar and Kumar, 2008; Devassy et al., 2021).

### Effectiveness of interventions by type and quality assessment

Most included studies ($n = 15$, 65.2%) reported a statistically significant positive impact of their intervention on suicide and/or NSSI-related outcomes.

Six of the eight studies evaluating interventions with predominantly psychotherapeutic content reported a significant positive effect (Agyapong et al., 2021; Puspitasari et al., 2021; Dias et al., 2023; Dominguez-Rodriguez et al., 2023; Obuobi-Donkor et al., 2024).

The highest quality studies examined *Text4Hope*, the CBT-informed texting service, which consistently reported reduced suicidal ideation and/or thoughts of NSSI after 6 weeks of daily text messages (Agyapong et al., 2021, 2023; Dias et al., 2023; Obuobi-Donkor et al., 2024); and the remote IOP prioritising DBT, BA therapy, and OT for high-risk individuals, which was associated with reductions in suicide risk (Puspitasari et al., 2021).

Two studies evaluated interventions drawing from multiple therapeutic models that were associated with statistically significant reductions in suicidal ideation and NSSI (Gliske et al., 2022; Stevens et al., 2022). The higher quality of these studies involved the remote IOP combining both psychotherapeutic and experiential approaches – with group DBT being provided to high-risk individuals (Gliske et al., 2022).

Of the five studies evaluating direct crisis management interventions, two in-person approaches – emotional support alone (Vijayakumar and Kumar, 2008) and emotional support with safety planning (Vijayakumar et al., 2017) – showed significant positive effects, though they were deemed to be of low quality. Among the skills-based interventions, both positive affect training (Bryant et al., 2023) and EI training (Persich et al., 2021) were associated with significant reductions in suicidal ideation. However, only the evaluation done by Bryant et al. (2023) was assessed as high-quality. The high-quality study done by Kelly et al. (2003) found that, among individuals aged 30 years and above, there was a significant negative association between the rate of prescription of antidepressant medication and recorded cases of suicide and undetermined deaths. In another high-quality study, Gujral et al. (2022) reported that the provision of video-enabled tablets led to a significant decrease in the likelihood of a suicide-related emergency department visit and the number of submitted SBORs. Finally, the study done by Abdulah and Abdulla (2020), of moderate quality, found that two months of creative arts therapy led to significant reductions in suicidal ideation.

## Discussion

This scoping review set out to synthesise the extant literature on interventions deployed in humanitarian settings to improve suicide and NSSI-related outcomes. A total of 23 articles were included, with most reporting positive effects of their interventions. However, multiple characteristics of these interventions necessitate nuanced discussion. Consistent with previous suicide prevention evidence syntheses from both humanitarian (Reifels et al., 2024) and non-humanitarian settings (Calear et al., 2016; Mann et al., 2021; Poudel et al., 2025), included articles varied in their quality and evaluated a heterogeneous pool of interventions – many of which involved multiple components, and relied, at least in part, on specialists for their implementation (see Table 2). The use of diverse, predominantly multicomponent, and specialist-led approaches is not surprising, given the variety of populations represented within included studies and the complex aetiology of suicide and NSSI (Knipe et al., 2022). However, challenges emerge when attempting to translate findings into actionable recommendations for humanitarian programming.

A principal challenge relates to the feasibility of implementing these interventions within the full breadth of contexts affected by humanitarian crises. The global impact of pandemics (i.e., COVID-19) notwithstanding, most humanitarian emergencies occur in LMICs (Al Omari et al., 2024) where both human and financial resources for mental health are scarce (Giebel et al., 2024). That

most interventions included in our review were implemented and evaluated in high-income countries (HICs), during the COVID-19 pandemic, and administered by specialists, reflects both previous reviews on humanitarian suicide prevention (Reifels et al., 2024) and the broader suicide-related literature, where <15% of research on suicide prevention takes place within LMICs (Knipe et al., 2022). Attempts to implement multicomponent interventions, particularly those relying on specialists for delivery, may therefore fall victim to a 'failure to launch' scenario, while high stigma, illegality of suicide, and the absence of national surveillance systems that capture data on suicide-related outcomes (WHO, 2025) present substantial barriers to sustainable implementation and scale-up (Barbui et al., 2020).

Beyond concerns regarding the feasibility of implementing interventions predominantly evaluated in HICs, there are similar uncertainties regarding the *applicability* of findings to LMICs, where the epidemiological profiles of individuals who die by suicide and/or engage in behaviours of self-harm – and the very conceptualisation of self-harm – may vary (Knipe et al., 2022). Taken together, this suggests an inadequate evidence base for effective suicide prevention strategies in LMICs (Knipe et al., 2022), and therefore, given their significant imbrication, humanitarian settings (Al Omari et al., 2024). Rectification of this knowledge gap requires urgent attention within humanitarian research efforts (Haroz et al., 2020; Reifels et al., 2024).

Despite these concerns, a subset of interventions stands out as promising opportunities to address the high risk for suicide and/or NSSI within humanitarian emergencies. The use of remote interventions for use in low-resource/humanitarian settings, particularly when considering issues of feasibility, accessibility (Ibragimov et al., 2022; Knipe et al., 2022), and scalability (Alvarez et al., 2022; He et al., 2023), for example, warrants further consideration.

Keeping in mind its self-subscription model (with results not necessarily reflective of individuals identified as high-risk for suicide), the CBT-informed automated texting service *Text4Hope* – which consistently demonstrated effectiveness in reducing suicide-related outcomes – stands out as particularly promising for reducing suicidal ideation and NSSI in an emergency with good mobile penetration and reliable coverage (Agyapong et al., 2021, 2023; Dias et al., 2023; Obuobi-Donkor et al., 2024). This finding is consistent with the broader scientific knowledge; in their *Lancet se*minar on suicide and self-harm, Knipe et al. (2022) assert that CBT-aligned approaches have the strongest evidence base for reducing suicidal ideation and repeat instances of self-harm. The many advantages of text-messaging services compared to more complex forms of remote health services (Ruzek and Yeager, 2017) – including well-documented cost effectiveness (Agyapong et al., 2023; Obuobi-Donkor et al., 2025) – together with its single-component approach and automated administration may help overcome both the stigma associated with seeking help and the limited number of human resources in humanitarian settings (Raftree, 2023; WHO, 2025). In addition, the significant increases in mobile phone ownership within low-resource settings (Maliwichi et al., 2024), including among displaced populations (Ashfaq et al., 2020), further highlight text-based CBT-aligned interventions as a promising suicide prevention intervention within humanitarian contexts. Incorporation of (an adapted) *Text4Hope* or similar programme into regional or national mental health policies – particularly those already engaging with digital health agendas – would likely benefit the intervention's efficient rollout following the onset of a humanitarian crisis (Agyapong et al., 2023; Obuobi-Donkor et al., 2024, 2025). Similarly, the leveraging of governmental early warning systems and/or mobile crisis information applications (Goniewicz and Burkle, 2019; Chan

and Tsai, 2023) may help facilitate timely and wide-reaching implementation of text-based mental health initiatives – keeping in mind the need for equitable access across affected populations (Goniewicz and Burkle, 2019).

Moreover, and consistent with the IASC's (2022) '*Addressing Suicide in Humanitarian Settings*', building life skills that serve as protective factors is an essential component of suicide prevention in humanitarian contexts. Two remote training programmes that made use of skills-based approaches – one targeting the general population (Persich et al., 2021) and the other individuals who screened positive for psychological distress (Bryant et al., 2023) – were associated with reductions in suicidal ideation. While the authors observed high participant drop-off, the positive effects of the EI training programme evaluated by Persich et al. (2021), for example, are consistent with previous meta-analyses and reviews recommending that EI training programmes be integrated into suicide prevention strategies (Domínguez-García and Fernández-Berrocal, 2018; Avanci et al., 2024; Darvishi et al., 2025). Like text-based services, its brief self-administered (more anonymous) nature may also be useful towards surmounting insufficient resources and significant stigma (Raftree, 2023; WHO, 2025), while also allowing for flexibility in user engagement (Raftree, 2023). However, the absence of a user-practitioner relationship likely implies that the usability of any self-administered programme is prioritised to support uptake and continuous use (Raftree, 2023). Similarly, the reduction in suicidal ideation associated with the brief positive affect training programme evaluated by Bryant et al. (2023) is corroborated by additional evidence (Bennardi et al., 2019; Teismann et al., 2019; Yen et al., 2020, 2023), suggesting that this may be another useful method of protecting against suicide and NSSI in humanitarian contexts. Importantly, Bryant et al. (2023) note that their positive affect intervention was delivered by clinical psychologists, emphasising how 'substantive scale-up…especially in low- and middle-income countries' will require the development of 'structured training protocols… for people with varying qualifications' (p. 6).

Implementing remote interventions, however, requires careful consideration to ensure their effectiveness and sustainability. These include community-driven cultural/contextual adaptations (IASC, 2022, Maliwichi et al., 2024); identification of logistical barriers (Komi et al., 2021), particularly regarding existing communications infrastructure (Ibragimov et al., 2022); ensuring inclusive service delivery (Komi et al., 2021; Maliwichi et al., 2024); and mitigating ethical challenges associated with data security (Komi et al., 2021; He et al., 2023). That said, Komi et al. (2021) and He et al. (2023) put forward useful conceptual frameworks for integrating remote initiatives into humanitarian response. Future implementation research on remote interventions – including documentation of context-specific adaptations (Reifels et al., 2024) and details on cost-effectiveness (Bowsher et al., 2021; Komi et al., 2021) – are required to advance the evidence base (Haroz et al., 2020; Reifels et al., 2024) and to develop standard protocols for delivering remote Mental Health and Psycho-Social Support (MHPSS) in humanitarian settings, as advocated for by Ahmed and Huen (2024).

While remote initiatives serve as an advantageous – and perhaps, as Komi et al. (2021) contend, necessary – component of humanitarian response, significant limitations to their wholesale implementation remain (Ibragimov et al., 2022; Parkes et al., 2022). In their guide on designing digital (i.e., remote) MHPSS interventions for displaced populations, the United Nations High Comissioner for Refugees (UNHCR) categorises these limitations into five areas: access and inclusion; relevance, trust,

and credibility; user context; digital protection; and a lack of evidence-based approaches (Raftree, 2023). Given their associated risks, some argue that the role of remote interventions should be to amplify, rather than substitute in-person service delivery (Armijos et al., 2023).

Two in-person interventions included in our review emerge as promising in this regard (Vijayakumar and Kumar, 2008; Vijayakumar et al., 2017), particularly given their administration by trained lay (non-specialist) individuals, as a well-established strategy to increase access to mental health services in contexts of low human resources (Knipe et al., 2022; Yankam et al., 2023). While assessed as low-quality, the *CASP* intervention, which focuses on providing regular emotional support and safety planning to individuals at high-risk of suicide, was found to reduce rates of suicide attempt and death by suicide (Vijayakumar et al., 2017) and is specifically mentioned within the IASC's (2022) '*Addressing Suicide in Humanitarian Settings*' guidance note. Similarly, the befriending intervention evaluated by Vijayakumar and Kumar (2008), which centres the provision of regular emotional support, was found to be associated with a reduction in suicide attempts over the course of the intervention's delivery and is consistent with creating a 'protective and supportive environment and a feeling of social connectedness' (IASC, 2022, p. 22). Indeed, the utility of these approaches is supported by robust evidence base. Multiple systematic reviews and meta-analyses highlight the effectiveness and feasibility of safety planning in suicide prevention among adult populations (Ferguson et al., 2021; Nuij et al., 2021; Marshall et al., 2022), highlighting its adaptability for individuals with distinct demographic profiles and support needs (Ferguson et al., 2021), with Rogers et al. (2022) cautioning against implementing safety planning as a standalone intervention. Meanwhile, the importance of promoting community and family cohesion is considered an integral component of protecting against mental distress within humanitarian crises (Miller et al., 2021; Papola et al., 2024).

Like replication of remote interventions, future implementation of these in-person interventions must undergo an assessment of their need for cultural adaptation (Jordans and Kohrt, 2020; Perera et al., 2020). Moreover, the use of lay individuals requires regular supportive supervision (IASC, 2007; Travers et al., 2022) of those directly responsible for intervention delivery. Designed specifically for individuals delivering MHPSS services in humanitarian settings, the 'Integrated Model for Supervision' (IFRC PS Centre and TCGH, 2023) offers useful guidance for how supervision can help protect the well-being and professional capacities of those delivering MHPSS (Ryan et al., 2025).

### *Future research directions*

Our results suggest several key areas for future research on suicide prevention in humanitarian emergencies. Principal among these is the dearth of research conducted in LMICs (Knipe et al., 2022). Given the disproportionate burden of suicide in these settings (WHO, 2025), future research on suicide aetiology, epidemiology and prevention in LMICs (Lovero et al., 2023) – including among populations affected by humanitarian crisis (IASC, 2022) – is not only an ethical imperative but is essential towards meeting global development goals (UN, 2025). While requiring careful navigation of the significant stigma and legal repercussions surrounding suicide in many contexts (Knipe et al., 2022; WHO, 2025), research is needed for the development of more robust global surveillance systems of suicide-related outcomes (IASC, 2022; Knipe et al., 2022). One potential avenue for this research is to investigate the

feasibility and utility of integrating a standalone indicator and means of verification (MoV) of suicide and NSSI risk within the IASC's (2021) guidance note on the monitoring and evaluation of humanitarian MHPSS programming. While critical for evaluating MHPSS activities in humanitarian settings, the lack of a suicide-specific MoV within this guidance note risks undermining its stated purpose of 'build[ing] the MHPSS evidence base and better inform[ing] those working in' (IASC, 2021, p. 12) humanitarian MHPSS – an aim that necessarily includes suicide prevention.

Additional routes for future research include the adaptation, replication and evaluation of the interventions highlighted in our review, as well as the evaluation of the downstream impact of health worker training interventions on beneficiary-level suicide-related outcomes. For instance, while mhGAP has been widely implemented across humanitarian settings (Humayun et al., 2017; Keynejad et al., 2021), investigations into whether and, if so, how its implementation translates into reduced rates of suicide and/or NSSI remain limited (Haroz et al., 2020).

Moreover, there is limited evaluative research done on suicide prevention for populations affected by armed conflicts, natural disasters or forced displacement (Knipe et al., 2022) – all of which are common (UNOCHA, 2024) and are likely characterised by a more complex constellation of suicide risk factors compared to COVID-19. Relatedly, there is minimal knowledge around effective interventions for suicide and/or NSSI prevention among specific at-risk sub-populations, including survivors of gender-based violence (Nam et al., 2023; Patel et al., 2024); persons with disabilities (Marlow et al., 2021; Koly et al., 2024); lesbian, gay, bisexual, transgender and queer/questioning individuals (Burgess et al., 2021; Paudel et al., 2024); and indigenous populations (Pollock et al., 2018), all of whom must be meaningfully involved in the development, delivery and research of suicide prevention interventions (Pollock et al., 2018; Burgess et al., 2021; IASC, 2022). Finally, while multisectoral approaches to suicide prevention are considered essential (IASC, 2022) – with combined systems-level approaches demonstrating effectiveness across multiple non-humanitarian settings (Mann et al., 2021) – there is a need to identify which combination(s) of intervention(s) are most effective in reducing the risk of suicide within humanitarian settings, as well as to clarify *how* and *when* they should be integrated into humanitarian programming.

### Strengths and limitations

This scoping review has several strengths. First, we focused on suicide and self-harm prevention across *all* types of humanitarian crises, thus differentiating our review from past similar efforts (Haroz et al., 2020; Reifels et al., 2024). Second, our adherence to the PRISMA-ScR checklist (Tricco et al., 2018) enhances the 'rigour, reproducibility and quality' of our review, thus improving its value and utility to end users (Peters et al., 2021, p. 4). This is a notable strength particularly when considering the proliferation of scoping reviews that fail to do so (Peters et al., 2021). Similarly, our use of a standardised tool to assess the methodological quality of each of our included studies allowed us to make more nuanced interpretations and thoughtful recommendations (Peters et al., 2021). Fourth, by placing no restrictions on the year or language of publication during the screening process, we were able to capture a wider range of potentially relevant records.

Our scoping review has three principal limitations. First, we excluded studies that evaluated higher-level suicide prevention interventions, such as governmental policies or restricting access

to lethal means (see Table 1). While we elected to do this to only capture interventions feasibly deliverable by humanitarian practitioners, it nonetheless ignores population-level strategies proven to be effective in preventing suicide (Hawton et al., 2024), including following humanitarian crises (see Matsubayashi and Kamada, 2021). Second, our review was not concerned with qualitative findings related to humanitarian suicide prevention activities. Due to our focus on *effectiveness* of interventions, this absence of qualitative evidence overlooks important dimensions related to the lived experience of those who engage in suicide prevention services (Watling et al., 2022), such as intervention acceptability, feasibility and participant-driven identification of barriers to access and areas for intervention improvement (Blattert et al., 2022; O'Brien et al., 2022; Castillo-Sánchez et al., 2024). Finally, about one-third ($n = 8$) of the studies included in our review were deemed to be of low quality, evincing a need for more high-quality research focused on the prevention of NSSI and suicide in humanitarian settings.

### Conclusion

As the number of individuals affected by armed conflict, natural disasters and forced displacement continues to grow (UNOCHA, 2024) – alongside the looming risk of future pandemics (Global Preparedness Monitoring Board, 2024) – the need for effective interventions to address the associated elevated risk of suicide and self-harm becomes increasingly urgent.

We conducted the first scoping review aimed at identifying and synthesising the extant literature on effective interventions for preventing suicide and/or self-harm across the entire spectrum of humanitarian and fragile contexts. We identified a selection of promising approaches, including CBT-based interventions, skills-building programmes that promote protective factors and strategies that foster a supportive and protective environment for high-risk individuals. Moreover, while acknowledging their limitations, we point to the potential of remotely administered interventions to augment the provision of in-person services. This becomes particularly important in LMIC settings, where most humanitarian crises occur.

Nevertheless, our findings point to a notable scarcity of literature in this area. Most studies originate from HICs, despite the disproportionate burden of both humanitarian crises and suicide in LMICs. This emphasises the resounding need for increased implementation and evaluative research of suicide prevention strategies in humanitarian settings – especially within lower resourced settings.

**Open peer review.** To view the open peer review materials for this article, please visit http://doi.org/10.1017/gmh.2025.10108.

**Supplementary material.** The supplementary material for this article can be found at http://doi.org/10.1017/gmh.2025.10108.

**Data availability statement.** Data sharing associated with this study is not applicable – no data were collected or generated as part of this scoping review.

**Acknowledgements.** The authors would like to thank the members of the Advisory Group for this project, who voluntarily lent their time to guide our efforts, including by supplying resources for the authors' grey literature review. The authors would also like to thank Mel Ó Súird for offering feedback on the structure and readability of the manuscript. Finally, the authors offer their sincerest of thanks to all those engaged in efforts of suicide and self-harm prevention – in both humanitarian and non-humanitarian contexts – across the globe.

During the preparation of this work, the author(s) used ChatGPT to improve the readability and language of certain excerpts of the manuscript.

After using this tool/service, the author(s) reviewed and edited the content as needed and take(s) full responsibility for the content of the published article.

**Author contribution. CZ:** Conceptualisation, methodology, data curation, investigation, formal analysis, validation, visualisation, project administration and writing – original draft. **FV:** Conceptualisation, methodology, investigation, project administration, supervision, funding acquisition and writing – review and editing. **FB:** Investigation, writing – original draft and writing – review and editing. **EEH:** Writing – review and editing. **IK:** Investigation, writing – original draft and writing – review and editing. **GS:** Methodology and writing – review and editing. **JSYL:** Writing – review and editing. **SH:** Project administration and supervision. **RS**: Conceptualisation, methodology, data curation, investigation, visualisation, project administration, writing – original draft and writing – review and editing. All authors approved the final version of the manuscript for publication.

**Financial support.** This work was supported by the American People through the United States Agency for International Development (USAID) (grant number 720BHA21IO00253). The contents of this study are the sole responsibility of the authors and do not necessarily reflect the views of USAID or the United States Government.

**Competing interests.** The authors declare none.

**Ethics statement.** All authors declare to adhere to the publishing ethics of Global Mental Health. Ethical approval for this study is not applicable – no primary data were collected, nor secondary data were analysed as part of this scoping review.

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
