## [Reviewer Report]

Overall, very well written. Future research directions given. In-depth scoping review of literature. Recognized limitations to this study.

Only a minor comment to address:

While not a limitation, 8/23 studies are below the 50% threshold based on your Mixed Methods Appraisal. Please mention this in your discussion or limitation. Because this is a scoping review would recommend leaving these in your study and addressing the potential lower quality of the studies.

Dr. Spencer Lord, MD, from Mass General Hospital in Boston, MA, reviewed this manuscript with me.

---

## [Reviewer Report]

Thank you for the opportunity to review this scoping review. Overall, it is a strong piece of research with some areas for potential improvement, which I have outlined below.

ABSTRACT

The abstract appears to be longer than the standard word limit for GMH. I recommend making it more focused on the primary research gap and study findings, rather than detailing specific methodological details.

INTRODUCTION

The sentence “Yet these gains remain inequitably distributed; low and middle income countries (LMICS) account for 73% of suicides” is not clearly linked to the preceding sentence. It suggests these gains have not occurred in these settings, but it may simply be that there were higher rates in the first place. It would be good to make this clearer.

METHODS

The review follows the SCR checklist, but it is not included with the manuscript. It would be good to include a completed version of the checklist as an appendix (https://www.equator-network.org/wp-content/uploads/2018/09/PRISMA-ScR-Fillable-Checklist.pdf)

The search strategy was comprehensive and well thought out. However, you may wish to consider adding a block of country names to your context search terms, using a recognised source such as the World Bank’s Fragile, Conflict and Violence (FCS) list or the UN OCHA Humanitarian Response Plan country list. This may improve recall by capturing studies that reference the country name without explicitly describing the setting as ‘humanitarian’ or ‘fragile.’ However, this will likely increase the number of records retrieved and may add to the screening burden, so the decision depends on your team’s capacity and your preference for sensitivity vs. specificity.

The screening process is clearly described and aligned with best practice. However, it is unclear how the two authors who completed the extraction addressed any discrepancies.

RESULTS

The results are well described. You may wish to add a paragraph summarising the quality of the available evidence.

DISCUSSION

Depending on how you decide to proceed with additional search terms, you may wish to note that you did not search for specific countries or known conflicts/fragile states with your search terms.

REFERENCES

I suggest adding an asterisk or other identifying factor to references included in the review in the reference list.

GENERAL ISSUES

The phrasing could be improved in some parts of the manuscript. For instance, “In addition to conflict and natural disasters, public health emergencies represent another form of crisis that can increase suicidal ideation (Cénat et al.,2020; Gunnell et al., 2020; Yan et al., 2023) - as another strong predictor of suicide (Reeves et al., 2022) - as well as NSSI (Farooq et al., 2021)”

---

## [Editor Report]

The authors present a strong scoping review for suicide and self-harm prevention in humanitarian and fragile settings. This is an important contribution to the literature and would benefit the wider space given the current situation in the MHPSS space. The reviewers have outlined key areas of improvement, please address these carefully. I have further comments of my own below:

1. Please outline which authors performed the screening

2. Please describe what search terms were used in the methods section briefly.

3. For Quality assessment, please briefly mention the criteria for quality using the MMAT

4. In the results section, please avoid using terms such as ‘small and moderate’ as these effect size categories are subjective and may not have utility in determining the real world effectiveness of these interventions

5. The discussion goes into depth of what is present and what is missing; however I think a deeper discussion into comparing the benefits and shortcomings of the different types of interventions would help readers, as well as stakeholders grasp the nature of the field, and move towards a mutual understanding of what to apply, when. 

6. Given that a meta analysis was not done, please elaborate on the interventions that are considered ‘promising’, specifically why these were considered ‘promising’. 

Thank you and all the best,

Dr. Sandersan Onie

---

## [Reviewer Report]

The authors have done a good job revising the article and responding to the reviewer comments. From my perspective, the article is appropriate for publication in its current form.

---

## [Editor Report]

Dear Prof Zemp,

Thank you for carefully addressing the comments posed by the reviewers. I am satisfied and am recommending this article for publication.

Thank you and all the best,

Dr Sandersan Onie